# Combinatorial Effects of Cisplatin and PARP Inhibitor Olaparib on Survival, Intestinal Integrity, and Microbiome Modulation in Murine Model

**DOI:** 10.3390/ijms26031191

**Published:** 2025-01-30

**Authors:** Mitsuki Matsumura, Hisako Fujihara, Kanna Maita, Moeko Miyakawa, Yushi Sakai, Ryoko Nakayama, Yumi Ito, Mitsuhiko Hasebe, Koji Kawaguchi, Yoshiki Hamada

**Affiliations:** 1Department of Oral and Maxillofacial Surgery, School of Dental Medicine, Tsurumi University, 2-1-3 Tsurumi, Tsurumi-ku, Yokohama 230-8501, Kanagawa, Japan; pd21010@stu.tsurumi-u.ac.jp (M.M.);; 2Department of Oral Hygiene, Tsurumi Junior College, 2-1-3 Tsurumi, Tsurumi-ku, Yokohama 230-8501, Kanagawa, Japan; 3Department of Pathology, School of Dental Medicine, Tsurumi University, 2-1-3 Tsurumi, Tsurumi-ku, Yokohama 230-8501, Kanagawa, Japan; 4Department of Diagnostic Pathology, Tsurumi University Dental Hospital, Yokohama 230-8501, Kanagawa, Japan

**Keywords:** PARP inhibitor, Olaparib, cisplatin, microbiome, pyroptosis, survival, dysbiosis

## Abstract

This study investigated the effects of the poly (ADP-ribose) polymerase (PARP) inhibitor Olaparib, alone and in combination with cisplatin, on intestinal integrity, survival, and microbiome composition using a murine model. Statistical analyses were conducted using one-way analysis of variance with Bonferroni correction for multiple comparisons, considering *p*-values of <0.05 as statistically significant. Microbiome profiling was performed using Qiime 2 software. Histopathological and microbiome analyses revealed Olaparib’s protective effects on intestinal integrity, mitigating cisplatin-induced damage. The single administration of cisplatin caused significant histological damage, biochemical disruptions, and dysbiosis, characterized by an increase in pro-inflammatory microbiome, such as *Clostridium_sensu_stricto_1*, and a decrease in beneficial short-chain fatty acid (SCFA)-producing microbiome. Conversely, the single administration of Olaparib was associated with an increase in SCFA-producing microbiome, such as *Lachnospiraceae NK4A136*, and exhibited minimal toxicity. The combination administration showed complicated outcomes, as follows: reduced cisplatin-induced cytotoxicity and increased SCFA-producing microbiome ratios, yet the long-term effects revealed reduced survival rates in the cisplatin group and sustained weight gain suppression. These findings emphasize Olaparib’s potential in enhancing intestinal barrier integrity, reducing inflammation, and positively modulating microbiome diversity. However, the entangled pharmacodynamic interactions in the combination administration underscore the need for further investigation. The study highlights the potential of microbiome-targeted interventions in improving therapeutic outcomes for both cancer treatment and inflammatory bowel disease management.

## 1. Introduction

In the last five years, poly (ADP-ribose) polymerase inhibitors (PARPis) have emerged as a pivotal targeted therapy for ovarian and breast cancers, particularly in patients with mutations in homologous recombination repair genes or deficiencies in DNA repair pathways, such as Brest Cancer Gene 1/2 (BRCA1/2) mutations [1,2]. Randomized clinical trials have demonstrated remarkable responses to PARPis in these cancers, leading to ongoing trials investigating their efficacy in endometrial and cervical cancers, as well as other non-gynecologic solid tumors [3]. Furthermore, findings from the phase III OReO/ENGOT Ov-38 trial (Olaparib Retreatment in Patients with Platinum-Sensitive Relapsed Ovarian Cancer/European Network for Gynaecological Oncological Trial Group Ov-38 trial) (NCT03106987) indicate that PARPi rechallenge provides a survival benefit in patients with platinum-sensitive relapsed ovarian cancer [4].

Despite their efficacy, adverse events associated with PARPis have been also reported, including hematologic toxicities (e.g., anemia, thrombocytopenia, and neutropenia), gastrointestinal symptoms (e.g., nausea, vomiting, and diarrhea), fatigue, renal dysfunction, and taste disorders. Data from the SOLO1 (Study of Olaparib in Patients With BRCA-Mutated Advanced Ovarian Cancer) and SOLO2 (Study of Olaparib in Patients With Platinum-Sensitive Relapsed Ovarian Cancer) trials underscore the prevalence of grade 1–2 gastrointestinal events such as nausea (73–77%), fatigue (62–63%), and diarrhea (32–40%), with anemia being the most common grade 3–4 toxicity (19%) [3,5]. To mitigate these effects, dose management strategies, including interruptions, dose reductions, and treatment discontinuations, have been implemented. Real-world data align with clinical trial findings, reporting similar incidences of fatigue, nausea, diarrhea, and anemia in patients receiving PARPi treatment [1,6,7]. Poly (ADP-ribose) polymerase (PARP) is a critical enzyme involved in the repair of single-strand DNA breaks [8]. PARPis block this enzymatic activity, leading to the accumulation of single-strand breaks, which ultimately transition into double-strand breaks (DSBs). These DSBs are typically repaired by the homologous recombination repair (HRR) pathway. However, in cancer cells with BRCA mutations or homologous recombination deficiency (HRD), the HRR pathway is impaired. The use of PARPis in these contexts exacerbates the inability to repair DNA damage, culminating in synthetic lethality and selective cancer cell death [9,10,11].

In our previous in vitro and in vivo studies on oral-cancer-derived cells conducted in 2016 [12], we demonstrated the following: (1) the PARPi Olaparib exhibited anti-tumor effects comparable to cisplatin in a xenograft model of oral cancer, and (2) combining Olaparib with cisplatin produced synergistic anti-tumor effects [12]. These findings suggest potential therapeutic applications of PARPis beyond ovarian and breast cancers, extending to oral cancer. However, adverse events associated with PARPis, including those from combination therapies with platinum-based agents, have only recently been documented in randomized trials and real-world settings [4]. The underlying mechanisms causing these adverse effects, with both the single administration of PARPis and combination regimens, remain poorly understood.

In the present study, we focused on the gastrointestinal adverse effects of PARPis and examined the effects of cisplatin and PARPis on the intestinal tract in a murine model. Specifically, we investigated the intestinal histopathology and microbiome alterations induced by PARPis and their combination with cisplatin. Our analyses utilized conventional histopathological methods and 16S rRNA sequencing to elucidate the impacts of these treatments on intestinal integrity and microbial composition.

## 2. Results

### 2.1. Body Weight Changes and Survival Rate

All mice exhibited a time-dependent increase in body weight throughout the experimental period. The body weight increases in the cisplatin and combination groups were significantly lower at four weeks after the completion of drug administration compared to the other two groups. However, in the following weeks, the cisplatin group showed a more significant increase in body weight, reaching non-significance compared to the control group (Figure 1a). Survival outcomes were monitored for six months following the completion of drug administration. Kaplan–Meier survival curves revealed no dropouts in the control and Olaparib groups. In contrast, the cisplatin group showed an early decrease in survival, followed by a later significant reduction in the survival of the combination group 120 days after drug administration. Finally, the cisplatin and combination groups showed decreased survival compared to the control and Olaparib groups (*p* < 0.05) (Figure 1b).

### 2.2. Results of Biochemical Blood Test

The biochemical blood test results demonstrated deviations from the reference values for creatinine (CRE), potassium (K), total bile acids (TBAs), total bilirubin (T-Bil), alanine aminotransferase (ALT), albumin/globulin ratio (A/G), and triglycerides (TG) at 1, 3, and 6 months after the completion of drug administration.

One month after the drug administration was completed, the CRE levels in the cisplatin group were significantly elevated compared to both the control and Olaparib groups. At three months after drug administration, the CRE levels in the Olaparib group were also significantly higher than those in the control group, exceeding the reference range. Notably, potassium levels dropped significantly below the normal range in the Olaparib group one month after drug administration, while the potassium levels in the other groups remained within normal limits. At one month after administration, the TBA levels in the cisplatin group were significantly increased compared to both the control and Olaparib groups. However, the combination group did not exhibit significant differences in their TBA levels compared to the control group. T-Bil levels were significantly reduced in the cisplatin group compared to the control group at one month after the completion of drug administration. While the combination group also demonstrated a reduction in T-Bil levels, this decrease did not reach statistical significance. The ALT levels in the control group were significantly higher than those in the other three groups. Additionally, the A/G ratio in the cisplatin group was significantly elevated compared to the control group one month after drug administration. Six months after drug administration, the Olaparib group exhibited significantly lower TG levels, falling below the reference range, in comparison to the cisplatin group. Additionally, the calcium levels (Ca) in the Olaparib group were below the reference range at one month after drug administration. Overall, the combination group demonstrated relatively improved liver and kidney function compared to the cisplatin group, although the glucose levels in the combination group were lower than those in the control group and below the reference range. By six months after drug administration, no significant differences were observed among the groups for any parameter (Figure 2).

### 2.3. Histopathological Changes

Microscopic analyses were performed to evaluate histopathological alterations in the intestinal tracts of the four groups of ICR female mice at 1, 3, and 6 months after the completion of drug administration. At each time point, a detailed assessment of the jejunal and ileal villi, including measurements of the villi length (Appendix A), thickness (Appendix A), and area (Appendix A), was conducted. In the jejunum, the cisplatin group exhibited significantly shorter villi lengths at 1 and 3 months after drug administration than the other three groups. Furthermore, the villi widths in this group were markedly narrower at 3 months after treatment. Notably, the villi area showed no significant differences across all groups and time points (Appendix A). In the ileum, the cisplatin group demonstrated consistently shorter villi lengths compared to the other three groups at all evaluated time points. However, the villi widths were similar across all groups throughout the entire duration of the study. The villi area in the cisplatin group was significantly smaller than that of the control group at 3 months after drug administration. At 6 months, the villi area in the cisplatin group was significantly reduced compared to both the control and Olaparib groups (Appendix A).

Microscopic examination under high magnification revealed significant vacuolar degeneration in both the jejunal and ileal villi of the cisplatin group throughout the experimental period. The severity of vacuolar degeneration was most pronounced at 1 month after drug administration and progressively diminished in a time-dependent manner. Vacuolar degeneration was also observed in the combination group, but was less severe than in the cisplatin group. Notably, this change rebounded with severity in the combination group at 6 months after drug administration (Figure 3).

### 2.4. Immunohistochemical Analysis of PARP-1, EGF, and E-Cadherin

Additionally, immunohistochemical analyses were conducted to assess the expressions of poly (ADP-ribose) polymerase-1 (PARP-1), Epidermal Growth Factor (EGF), and E-cadherin to further investigate the atrophic and vacuolar changes observed in the intestinal villi.

#### 2.4.1. PARP-1 Expression

PARP-1 exhibited a significantly strong positivity not only in the surface mucosa of the jejunum, ileum, and colon, but also within cellular structures inside the villi in the cisplatin and combination groups at 1, 3, and 6 months after drug administration compared to the control and Olaparib groups. The quantification of DAB staining, converted to grayscale, revealed significantly lower grayscale values in the cisplatin and combination groups, which indicated significant positivity compared to the other groups. Notably, at 3 and 6 months in the combination group, Paneth cells displayed a markedly strong PARP-1 positivity (Figure 4 and Appendix A).

#### 2.4.2. EGF Expression

EGF expression was strongly positive in the surface mucosa of the jejunum, ileum, and colon in the control and Olaparib groups at 1 month after drug administration compared to the cisplatin and combination groups (Figure 5a). The Olaparib group exhibited a significantly higher EGF positivity than the control group (Figure 5b). However, this difference became inconsistent at 3 and 6 months after drug administration, showing no clear pattern across the groups or drug administration regimes (Appendix A).

#### 2.4.3. E-Cadherin Expression

E-cadherin demonstrated a strong positivity on the jejunal, ileal, and colonic mucosa surfaces in the cisplatin and combination groups at 1 month after drug administration (Figure 6). This pattern of expression persisted consistently throughout the experimental period (Appendix A). The expression profile of E-cadherin closely resembled that of PARP-1.

### 2.5. The Result of Microbiome Analysis

The sequencing depth of the raw data received from Genome-Lead Co., Ltd. (Takamatsu, Japan) consisted of both forward and reverse reads. The minimum number of reads per sample was 57,205 bp, the mean number of reads was 95,680.89 bp, and the maximum number of reads was 137,920 bp. The cluster density was 775 K/mm^2^, with 16.34% Phix control reads. In total, 94.82% of the reads passed quality filtering and the Q30 score was 82.87%, demonstrating a high accuracy in base calling and enough quality for further analysis.

#### 2.5.1. Within-Subject α-Diversity and Between-Subject β-Diversity After 1, 3, and 6 Months After Drug Administration

No significant differences in α-diversity were observed across all groups at 1 month after drug administration. However, a significant difference emerged between the control and combination groups at 3 months after drug administration, as well as between the cisplatin and Olaparib groups and the combination and Olaparib groups at 6 months after drug administration (*p* < 0.05) (Figure 7).

Regarding β-diversity, the cisplatin group exhibited significantly altered microbial community structures compared to both the control and Olaparib groups at 1 month after drug administration. At 3 months, combination group showed significant differences in microbial β-diversity compared to each of the other three groups (*p* < 0.05). By 6 months, the cisplatin group demonstrated a significant β-diversity difference relative to the Olaparib group (*p* < 0.05; Figure 8).

#### 2.5.2. Microbial Taxa Analysis at Phylum and Genus Level

The number of microbial taxa with relative abundances exceeding 5% and 0.1% was comparable across the four experimental groups, with no statistically significant differences detected (Appendix A).

At the phylum level, Firmicutes was identified as the predominant phylum, followed by Bacteroidota, collectively accounting for over 90% of the intestinal microbiome in all groups throughout the experimental period. Notably, the cisplatin group exhibited a significantly higher abundance of Firmicutes and a corresponding decrease in Bacteroidota at one month after drug administration. Additionally, Proteobacteria showed an increased relative abundance in both the cisplatin and combination groups at 1 and 3 months following drug administration (Figure 9a).

At the genus level, the distinct microbial taxa were as follows:f_*Muribaculaceae;* g_*Muribaculaceae*: This genus consistently represented approximately 20% of the microbiome across all experimental groups and time points. No significant differences in abundance were observed between the groups at any time point.f_*Lachnospiraceae*; g_*Lachnospiraceae_NK4A136_group*: Members of the *Lachnospiraceae* family demonstrated dynamic changes in abundance. At one month following drug administration, the relative abundances of the control, Olaparib, cisplatin, and combination groups were 5.83%, 6.18%, 1.98%, and 4.00%, respectively, with a statistically significant reduction observed in the cisplatin group (*p* < 0.05). By three months, these differences were no longer significant. Notably, at 6 months after drug administration, the relative abundances were 6.53%, 9.92%, 3.47%, and 8.90%, respectively, showing a significant decrease in the cisplatin group (*p* < 0.05) and a similar trend for the Olaparib and combination groups.f_*Clostridiaceae*; g_*Clostridium_sensu_stricto_1*: This genus exhibited marked group-specific abundance patterns. At 1 month after drug administration, its mean relative abundance was significantly higher in the cisplatin (16.78%) and combination (6.62%) groups compared to the control (0.23%) and Olaparib (0.00%) groups (*p* < 0.01) (Figure 9b).

#### 2.5.3. Microbiome Alterations Induced by Cisplatin and PARPis Identified by LEfSe Analysis

We applied the LEfSe algorithm to identify characteristic bacterial taxa where the abundance was significantly affected by the drug administration of cisplatin and Olaparib. At the taxonomic level six (genus level), the taxa with an LDA score exceeding four were as follows:

One month after drug administration, the cisplatin group exhibited a distinct microbiome characterized by a significant increase in the relative abundance of o_*Clostridiales;* f_*Clostridiaceae;* g_*Clostridium_sensu_stricto_1*. The Olaparib group was associated with a high LDA score for f_*Lachnospiraceae;* g_*Lachnospiraceae_NK4A136_group*. Three months after drug administration, the Olaparib group continued to show a high LDA score for f_*Lachnospiraceae; g_Lachnospiraceae_NK4A136_group*. In contrast, the control group exhibited an increased relative abundance of g_*Alloprevotella*. Six months after drug administration, only the Olaparib group demonstrated a sustainably distinct microbiome, characterized by an LDA score exceeding four for f_*Lachnospiraceae;* g_*Lachnospiraceae_NK4A136_group* (Figure 10 and Appendix A).

#### 2.5.4. Heatmap Analysis

We performed a heatmap analysis to visualize the relative abundances of the microbiomes across the four groups. The results were almost consistent with those obtained from the LEfSe analysis.

One month after drug administration, increased abundances of f_*Clostridiaceae*; g_*Clostridium_sensu_stricto_1* and f_*Erysipelotrichaceae*; g_*Turicibacter* were observed in the cisplatin and combination groups, with the cisplatin group showing a higher abundance compared to the combination group. In contrast, f_*Lachnospiraceae*; g_*Lachnospiraceae_NK4A136_group*, f_*Oscillospiraceae*; g_*Oscillibacter*, f_*Butyricicoccaceae*; g_*Butyricicoccus*, and f_*Lachnospiraceae*; g_*Marvinbryantia* were more prevalent in the control and Olaparib groups. Additionally, f_*Lachnospiraceae*; g_*ASF*356 and f_*Ruminococcaceae*; g_*Incertae_Sedis* were more abundant in the control group, while f_*Clostridia_UCG014*; g_*Clostridia_UCG014* was less abundant in the Olaparib group. Moreover, f_*Lachnospiraceae*; g_*Anaerostipes* was not detected in the control and Olaparib groups, but was slightly present in the combination group (Figure 11a and Appendix A). Three months after drug administration, no microbiome showed an increased abundance, although g_*Alloprevotella* and g_*Lachnospiraceae_NK4A136_group* showed a high LDA score exceeding four (Figure 11b and Appendix A). Six months after drug administration, f_ *Erysipelotrichaceae*; g_*Turicibacter* was absent only in the control group, but was observed in the other three groups. A decrease in f_*Lachnospiraceae*; g_*Lachnospiraceae_NK4A136_group* was observed in the cisplatin group, though it remained relatively detectable in other three groups (Figure 11c and Appendix A).

#### 2.5.5. Analysis of Composition of Microbiomes (ANCOM)

We performed Analysis of Composition of Microbiomes (ANCOM) to identify the microbial taxa that were significantly different in abundance within the four groups. g_*Lachnospiraceae_NK4A136_group* at 1 month after medication was only the taxa which showed a W statistic of 74.

## 3. Discussion

In the present study, we investigated the effects of the PARPi, Olaparib, both alone and in combination with cisplatin, on intestinal integrity, survival, biochemical blood parameters, and microbiome composition using female ICR mice. Using histopathological, biochemical, and microbiome analyses, we aimed to elucidate the therapeutic potential and limitations of these medications, particularly in mitigating chemotherapy-induced damage to intestinal conditions. The findings obtained from our multifaceted analyses revealed not only potential therapeutic benefits of Olaparib in reducing inflammatory damage, but also complex interactions between these drugs, underscoring the importance of optimizing combination therapies to maximize clinical benefits while minimizing adverse effects.

Cisplatin, a pivotal agent in chemotherapy protocols for a wide range of malignancies, is well-documented for its substantial systemic toxicity [13,14]. Our findings corroborate this, demonstrating that the single administration of cisplatin was associated with early mortality, weight loss, liver and kidney dysfunction, and marked intestinal damage. These effects were consistent with our histopathological findings and supported by the severity of vacuolar changes in the intestinal villi, as well as increased PARP-1 activity, indicating DNA damage and proinflammatory signaling [11]. These effects were also accompanied by a distinct shift in microbiome composition, suggesting a proinflammatory state. In contrast, Olaparib monotherapy demonstrated minimal adverse effects, with survival and weight outcomes comparable to the control group. This observation supports the notion that PARPis may exert protective effects on intestinal integrity, potentially mediated by anti-inflammatory pathways.

The combination treatment, however, presented a more nuanced outcome. While we initially hypothesized that Olaparib would mitigate cisplatin-induced toxicity, the combination group exhibited the slowest weight gain and a late-stage decline in survival, suggesting complex pharmacodynamic interactions between Olaparib and cisplatin. Interestingly, the cisplatin group demonstrated subsequent weight recovery, achieving a comparable weight to the control group by the end of the study. This recovery could be attributed to selective survival, where more resilient individuals survived the initial toxicity. The mechanisms underlying survival following intensive chemotherapy warrant further investigation, as the current literature primarily consists of narrative reviews [15].

Biochemical analyses revealed further insights into the systemic effects of the treatments. The cisplatin group showed significant alterations in renal and hepatic function parameters, including elevated CRE, TBA, and A/G levels and decreased T-BiL levels at 1 month after drug administration, consistent with its known nephrotoxicity and hepatotoxicity [13,14]. Moreover, Olaparib monotherapy resulted in a delayed and milder renal dysfunction, and the combination group showed decreased glucose levels, correlating with a reduced weight gain. These findings suggest that Olaparib may offer a safer alternative to cisplatin, although its potential effects on liver function should be monitored considering decreased alanine aminotransferase (ALT) levels.

Histopathological analysis revealed the ileum as being particularly susceptible to drug-induced damage, with significant reductions in villus length in both the cisplatin and combination groups. Although partial recovery was observed at six months, villus length remained significantly reduced compared to controls, indicating persistent damage. These groups also exhibited vacuolar changes indicative of cisplatin-induced pyroptosis [16,17], a form of programmed cell death that can trigger inflammation and contribute to chronic intestinal inflammation and sustained dysbiosis [18]. While the combination therapy initially attenuated cisplatin-induced villus damage, this protective effect diminished over time, with a subsequent increase in pyroptosis, suggesting complex pharmacodynamics with prolonged combination therapy.

Immunohistochemical analysis revealed an increased PARP expression in both the cisplatin and combination groups, reflecting the DNA damage response triggered by cisplatin [19]. However, the elevated PARP expression in the combination group was unexpected, given the anticipated inhibitory effects of Olaparib. This observation suggests that cisplatin might interfere with Olaparib’s ability to inhibit PARP, or that alternative pathways are involved in PARP activation in this context. The reduced EGF expression in the cisplatin and combination groups at 1 month after drug administration aligned with cisplatin-induced cytotoxicity, which disrupts epithelial barrier function [20,21] and downregulates EGF expression in the intestinal villi [22]. Given EGF’s role in the epithelial–mesenchymal transition (EMT) [23] and Olaparib’s known effects on EMT [24,25], the observed effects appeared to depend on the drug combination and timing.

Unexpectedly, E-cadherin expression, crucial for maintaining epithelial integrity, was increased in both the cisplatin and combination groups, contrary to typical cisplatin effects [26] and the expected effects of PARP-1 silencing [27]. Thus, we consider the following two plausible candidate pathways: the (1) β-catenin and (2) p53 pathways. PARP activation appears to exert context-dependent effects, either promoting or inhibiting β-catenin expression. PARP-1 expression correlates with β-catenin levels, and its inhibition has been shown to enhance cisplatin-induced cytotoxicity in cervical cancer cells via the modulation of the β-catenin signaling pathway [28]. Furthermore, β-catenin directly binds to the C-terminal catenin domain (CBD) of E-cadherin, thereby enhancing its function [29]. In the p53 pathway, PARP-1 overexpression has been reported to interfere with double-strand break (DSB) repair, independent of its enzymatic activity and poly (ADP-ribosyl)ation, potentially activating p53 [30]. Then, the upregulation of p53 plays critical role in E-cadherin expression by repressing ZEB1 expression [31].

Microbiome analyses revealed distinct profiles across groups. The control group maintained a balanced microbiome with a diverse representation of SCFA-producing genera, including *ASF*356 [32], *Incertae_Sedis* [33], and *Eubacterium ventriosum* [34], which support intestinal health and reduce inflammation. The Olaparib group showed consistently high levels of *Lachnospiraceae_NK4A136_group* [35,36], a beneficial SCFA producer, consistent with the microbiome observed in PARP KO mice [37,38]. This group also showed an increased abundance of other SCFA producers such as *Anaeroplasm a* [39] and *Roseburia* [40], and a decrease in the depression-related *Anaerostipes* [41], suggesting potential effects of Olaparib on promoting intestinal integrity [42] and mental health in addition to cancer treatment [43]. Notably, *Lachnospiraceae_NK4A136_group* was consistently identified across all analyses, providing strong evidence of Olaparib-induced microbiome alterations.

In contrast, the cisplatin and combination groups demonstrated significant dysbiosis, with an increased proinflammatory microbiome, including *Clostridium_sensu_stricto_1* [44] and *Turicibacter* [45,46], and a reduced SCFA-producing microbiome. The combination group, however, showed a dynamic microbiome profile, transitioning from a pro-inflammatory state to the later emergence of beneficial SCFA producers (e.g., lower ratio of *Clostridium_sensu_stricto_1* and *Turicibacter* and increased *Lachnospiraceae NK4A136*, *Oscillibacter* [47], *Butyricicoccus* [48], and *Marvinbryantia* [49]). These data suggest a partial mitigation of cisplatin’s adverse effects on the microbiome with the combination therapy, although the protective effects of Olaparib appeared attenuated in the presence of cisplatin.

This study has limitations that should be mentioned. First, hematological assessments were not feasible due to sample volume constraints. Consequently, the common side effects of PARP inhibitors, such as anemia and neutropenia [3,5], could not be evaluated. Second, the analysis of individuals that were removed from the study due to mortality was not possible. While investigating these individuals could have provided valuable insights into the factors directly related to mortality, the rapid postmortem degradation of intestinal tissue precluded histopathological and microbiome analyses. If the scheduled analyses of surviving individuals yield data comparable to those of the mortal individuals, the early mortality observed in the cisplatin group might have correlated with an increase in *Clostridium sensu stricto 1*, while the later mortality in the combination group might be linked to a higher expression level of PARP and a rebound of pyroptosis, as suggested by the histopathological analysis. Future studies should prioritize comprehensive hematological analyses and explore alternative approaches, such as immediate tissue collection or preservation techniques, to enable the evaluation of deceased animals and provide a more complete assessment of systemic toxicity. Thirdly, while 16S rRNA analysis provides a broader overview of the microbial community composition, its accuracy is comparatively lower than that of real-time PCR analysis. Therefore, further investigations, including precise quantification and the detailed exploration of associations with other factors, would be required.

In conclusion, this study provides critical insights into the effects of Olaparib and cisplatin on survival, biochemical blood parameters, and intestinal integrity. While the single administration of Olaparib demonstrated protective effects and minimal toxicity, its combination with cisplatin demonstrated complicated and nuanced results with a decreased cisplatin cytotoxicity, increased beneficial microbiome, and similar survival ratio with the cisplatin group (Figure 12). The impact of these treatments on the microbiome composition and intestinal histopathological changes further highlights the potential of microbiome-targeted interventions to improve therapeutic outcomes. By increasing SCFA-producing microbiota and reducing inflammatory and apoptotic changes, Olaparib may serve as a promising adjunct in both cancer treatment and the management of inflammatory bowel disease. Therefore, its role in combination regimens requires careful evaluation to mitigate long-term risks and maximize clinical benefits.

## 4. Materials and Methods

### 4.1. Animal Models

All animal experimental protocols were reviewed and approved by the Institutional Animal Care and Use Committee of Tsurumi University School of Dental Medicine, following the institution’s guidelines for the care and use of laboratory animals (Approval Number: 22P009, June 2022). This study strictly adhered to the ethical standards established for animal research.

Six-week-old female ICR mice (SLC Co., Ltd., Shizuoka, Japan), with average body weight of 29.2 g, were used for this study. The animals were housed under specific pathogen-free conditions throughout the experimental period and maintained at a controlled temperature (22–24 °C) with 12 h light/dark cycle. Mice were provided with unrestricted access to standard laboratory chow (CE-2; SLC Co., Shizuoka, Japan) and sterilized water to ensure the optimal nutrition and hydration.

To minimize animal suffering, humane endpoints were predefined to address any potential distress. The health and behavior of the animals were closely monitored daily, including observations of significant weight loss (>20% of baseline), unkempt fur, hunched posture, diminished mobility, or aberrant feeding behaviors. A system for rapid intervention was implemented, and humane euthanasia was set to be carried out in cases where severe distress or unresponsiveness to supportive measures was observed. All monitoring and interventions were conducted by trained personnel proficient in recognizing clinical indicators of discomfort, ensuring ethical and compassionate animal care throughout the experiments.

### 4.2. Procedure for Animal Experiments

One week following their acclimatization at our institution, the mice were randomly assigned to one of the following four experimental groups: (a) control (200 µL saline), (b) Olaparib (AZD2281, ChemScene, Monmouth Junction, NJ, USA, 25 mg/kg body weight, dissolved in 200 µL sterilized water), (c) cisplatin (cis-diammineplatinum (II) dichloride, Sigma-Aldrich Co. LLC, St. Louis, MO, USA, 2 mg/kg body weight, dissolved in 200 µL sterilized water), or (d) combination (cisplatin and Olaparib, with a total volume of 200 µL), maintaining consistency with our previously published protocol [12]. All compounds were administered via intraperitoneal injection every other day for a total of 10 doses. Although Olaparib (AZD2281) is typically administered orally in clinical practice, intraperitoneal injection was employed in this study based on the manufacturer’s recommendation to facilitate consistent dosing and avoid the potential stress and ethical concerns associated with oral gavage in mice. Following the completion of the 10-dose drug administration, the mice were randomly selected for euthanasia at 1, 3, and 6 months after the completion of drug administration. Euthanasia by cervical dislocation was performed under anesthesia induced by the intraperitoneal injection of a combination of medetomidine hydrochloride (Nippon Zenyaku Kogyo Co., Fukushima, Japan), midazolam (Astellas Pharma Inc., Tokyo, Japan), and butorphanol (Meiji Seika Pharma Co., Tokyo, Japan) [50]. Details of the sample sizes for each experimental group are summarized in Table 1.

### 4.3. Measurement of Body Weight and Survival Ratio

The body weight of all mice was measured weekly using an electronic animal balance scale (Entris II, Sartorius, Göttingen, Germany) to monitor changes over time and assess potential drug-related toxicity.

Kaplan–Meier survival curves were constructed to evaluate the survival rates of each experimental group, and survival data were statistically analyzed with the log-rank test to identify significant differences between groups. Mice that died during the observation period were recorded as dropouts from the survival analysis. The remaining animals were humanely euthanized at predetermined time points (1, 3, and 6 months after drug administration) for subsequent analyses.

### 4.4. Preparation of Blood Samples and Biochemical Blood Test

At 1, 3, and 6 months following drug administration, the mice were fully anesthetized, as described in our prior methodology [50]. Blood samples were then collected from the orbital sinus into 1.5 mL sterilized tubes [51]. The collected blood samples were allowed to clot for 30 min at room temperature and subsequently centrifuged at 1500× *g* for 15 min using a hybrid high-speed cooling centrifuge (KUBOTA 6200, Kubota Co., Ltd., Tokyo, Japan). The supernatant was carefully transferred to fresh 1.5 mL tubes, and the serum was stored at −20 °C until further analysis.

Biochemical blood tests of the collected serum were performed at Nagahama Life Science Laboratory (Oriental Yeast Co., Ltd., Shiga, Japan). The parameters analyzed were as follows: total protein (TP), albumin (ALB), albumin/globulin ratio (A/G ratio), blood urea nitrogen (BUN), creatinine (CRE), sodium (Na), potassium (K), chloride (Cl), calcium (Ca), inorganic phosphorus (IP), aspartate aminotransferase (AST), alanine aminotransferase (ALT), alkaline phosphatase (ALP), lactate dehydrogenase (LDH), amylase (AMY), gamma-glutamyl transferase (γ-GT), lipase (Lip), total cholesterol (T-CHO), triglycerides (TG), high-density lipoprotein cholesterol (HDL-C), total bilirubin (T-BIL), total bile acids (TBAs), and glucose (GLU).

### 4.5. Preparation of Fecal and Intestinal Samples

Following blood collection, fecal samples were promptly collected, followed by being frozen and stored at −20 °C for subsequent analysis [52]. The entire digestive tract, including the esophagus, stomach, duodenum, jejunum, ileum, cecum, and colon, was then harvested and processed using the Swiss-rolling technique [53,54]. The procedure began with an incision in the abdominal skin, allowing for careful excision of the entire intestine from the distal rectum/anus junction to its connection with cecum. The colon was immediately opened longitudinally to collect fecal samples. The remaining sections of the digestive tract were carefully separated from the surrounding mesenteric connective and adipose tissue, longitudinally incised along the mesenteric border, and rinsed thoroughly with phosphate-buffered saline (PBS) to remove residual contents. For tissue processing, the incised digestive tract was laid flat on a rubber plate, with the luminal surface facing upward. The tissues were gently pinned to maintain alignment and fixed with 10% formaldehyde. The fixed tract was then rolled from the proximal end of the stomach to the cecum and colon using the Swiss-rolling technique. Finally, the rolled tissues were placed in 10% neutral buffered formalin solution (FUJIFILM Wako Pure Chemical Corporation, Osaka, Japan) and stored at 4 °C for 48 h before further analysis.

### 4.6. Histopathological Analysis—Villi Length, Thickness, and Area

After fixation, the samples were immersed in 70% alcohol for 48 h and subsequently embedded in paraffin (SAKURA Tissue-Tek, Sakura Finetek Japan, Tokyo, Japan). Tissue sections were cut a 5 µm thickness, mounted on silane-coated slides (New Silane III, Muto Pure Chemicals Co., Ltd., Tokyo, Japan), deparaffinized with xylene, and rehydrated through graded ethanol series (FUJIFILM Wako Pure Chemical Corporation, Osaka, Japan). Hematoxylin and eosin (H&E) staining (Merck KGaA, Darmstadt, Germany) was performed to visualize tissue morphology. The stained sections were analyzed under a BX51 System Microscope (Olympus, Tokyo, Japan), and high-resolution images were captured using a DP70 digital microscope camera equipped with Mosaic 2.4 software (Bio Tools K.K., Gunma, Japan). To assess structural alterations, the villi length, thickness, and area were measured in ten randomly selected regions within the jejunum and ileum of each section. The image analysis was conducted using the ImageJ software, version 1.54f (National Cancer Institute, Bethesda, MD, USA). Statistical comparisons of the villi length, thickness, and area were performed across the experimental groups at 1, 3, and 6 months after drug administration.

### 4.7. Immunohistochemical Analysis and Its Semi-Quantitative Analysis

Following deparaffinization with xylene and rehydration through a graded series of ethanol concentrations, antigen retrieval was performed using Immunosaver (Nisshin EM, Tokyo, Japan) according to the manufacturer’s instructions. Briefly, the sections were incubated in Immunosaver (1:200 dilution in tap water) for 40 min at 95 °C and transferred to tap water for incubation for 10 min at room temperature. Endogenous peroxidase activity was subsequently inactivated by treating the slides with 3% hydrogen peroxide in methanol (both from Fujifilm Wako Pure Chemical Corporation, Osaka, Japan) for 30 min at room temperature, followed by blocking with 20% normal goat serum (Nichirei Corporation, Tokyo, Japan) for 30 min at room temperature. The sections were incubated with the primary antibodies overnight at 4 °C. The primary antibodies used in this study were as follows: anti-poly (ADP-ribose) polymerase (PARP) antibody (1:400; Abcam, Cambridge, UK), anti-epidermal growth factor (EGF) antibody (1:400; Bioss, Woburn, MA, USA), and anti-E-cadherin antibody (1:400; Gene Tex, Irvine, CA, USA). The antibodies were diluted in PBS (pH 7.4) containing 1% bovine serum albumin (Sigma-Aldrich, St. Louis, MO, USA) and incubated at 4 °C. After multiple PBS washes, the bound antibodies were visualized using Histofine Simple Stain MAX-PO (Rabbit) (Nichirei Corporation, Tokyo, Japan) and DAB (Vector Laboratories, Burlingame, CA, USA), according to the manufacturer’s instructions. The sections were subsequently counterstained with hematoxylin. For negative controls, sections were processed under the same conditions without exposure the primary antibodies. 

Digital images of the stained sections were captured and analyzed semi-quantitatively using the ImageJ software with the Color Deconvolution plugin (NCI, Bethesda, MD, USA) [55]. The “H DAB” option within the Color Deconvolution plugin was employed to isolate the DAB (3,3′-diaminobenzidine) channel, enhancing the specificity of the staining analysis. The resulting DAB channel images were converted to 8-bit grayscale and subjected to thresholding to delineate positively stained regions. Quantification was performed using the “Analyze Particles” function in ImageJ, enabling the calculation of the percentage of positively stained area and the integrated density (area × mean gray value) for each image. The staining intensity was averaged across multiple representative fields for each sample to ensure a robust and semi-quantitative evaluation of antigen expression.

### 4.8. DNA Extraction and 16S rRNA Gene Amplicon Sequencing

#### 4.8.1. DNA Extraction from Fecal Samples

The total genomic DNA was extracted from the fecal samples using the ISOSPIN Fecal DNA Kit (Nippon Gene Co., Ltd., Tokyo, Japan), following the manufacturer’s protocol. The quality and concentration of the extracted DNA were assessed using a NanoDrop One spectrophotometer (Thermo Fisher Scientific Inc., Waltham, MA, USA). DNA integrity was verified through 1% agarose gel electrophoresis to ensure suitability for downstream applications.

#### 4.8.2. 16S rRNA Gene Amplification and Library Preparation

The amplification and library preparation were outsourced to Genome-Lead Co., Ltd. (Takamatsu, Kagawa, Japan). To profile the microbial communities, V3–V4 hypervariable regions of the bacterial 16S rRNA gene were targeted. A dual-indexing strategy was employed, using primers containing unique 17- or 21-base adapters to facilitate sample multiplexing and compatibility with Illumina sequencing. The primers used were as follows:Forward Primer (F): TCGTCGGCAGCGTCAGATGTGTATAAGAGACAGNCCTACGGGNGGCWGCAGReverse Primer (R): GTCTCGTGGGCTCGGAGATGTGTATAAGAGACAGNGACTACHVGGG-TATCTAATCC

Initial PCR amplification was conducted to amplify the V3–V4 regions from each sample. This was followed by a second PCR step incorporating adapter and index sequences to enable sample demultiplexing and pooling into a single library format [56,57,58]. Sequencing was also performed by Genome-Lead Co., Ltd. using the Illumina MiSeq platform (Illumina Inc., San Diego, CA, USA). A paired-end 2 × 301 bp sequencing run was utilized to generate high-quality reads for downstream analyses.

### 4.9. Analysis of Experimental Data

#### 4.9.1. Microbiome Data

After we received the raw data, their quality was checked with sequencing depth, cluster density, Phix, filtering ratio, and Q30 score. Then, the detailed microbiome composition was obtained by uploading these raw data on QIIME2 v2021.4 [52]. Taxonomic assignment was performed against a reference database (Silva 128 SEPP reference database, MD5: 7879792a6f42c5325531de9866f5c4de) to profile the microbial communities.

#### 4.9.2. Procedure of Microbiome Analysis

The indices of the microbiome analysis were as follows: α-diversity, β-diversity, the microbiome composition ratio at level 2 and 6, linear discriminant analysis (LDA) effect size (LEfSe), heatmap analysis, and Analysis of Composition of Microbiomes (ANCOM).

The within-subject α-diversity of the bacterial communities was assessed using the Shannon index and the observed number of operational taxonomic units (OTUs). These indices were compared among the four groups using the Kruskal–Wallis test. Between-subjects β-diversity was evaluated based on Bray–Curtis dissimilarity and unweighted and weighted UniFrac distance metrics. Principal coordinate analysis (PCoA) was performed to visualize global differences in the microbiome structures based on UniFrac metrics. The statistical significance of the compositional differences between groups was assessed by permutational multivariate analysis of variance (PERMANOVA). These analyses were conducted using QIIME2 v2021.4 [52,59,60,61].

Differentially abundant bacterial genera among the four groups were identified using the linear discriminant analysis (LDA) effect size (LEfSe) algorithm. LEfSe’s α parameter for pairwise tests was set to <0.05, and the logarithmic score threshold for LDA was set to >4.0. Following model construction, the variable importance in projection (VIP) of each level (from phylum to species) was calculated, and the candidate genera were selected to highlight significant differences among the groups [62,63]. *p*-values of <0.05 were considered to be statistically significant. Heatmap analysis was conducted to visualize the differential abundances of microbial taxa across the groups at various taxonomic levels, from phylum to species, using hierarchical clustering to illustrate group-specific patterns [64,65]. Analysis of Composition of Microbiomes (ANCOM) was performed to identify microbial taxa with significantly different abundances between sample groups via a plugin on QIIME2 v2021.4 [66]. ANCOM accounts for the compositional nature of microbiome data by performing pairwise comparisons of each taxon against all others, generating a W statistic indicating the number of times a taxon is differentially abundant. A W statistic of 20 or greater, along with a “TRUE” designation in the corresponding table, was used as a threshold for identifying taxa with a significant differential abundance between groups.

#### 4.9.3. Statistical Analysis

The data collected from the four experimental groups, including measurements of body weight, survival ratio, biochemical blood test data, parameters of villi morphology, semi-quantitative immunohistochemistry (IHC) analysis, and microbiome composition ratios, were analyzed using the EZR software, version 1.62 (Saitama Medical Center, Jichi Medical University, Saitama, Japan), a graphical user interface for R (The R Foundation for Statistical Computing, Vienna, Austria). This modified version of R Commander is designed to add statistical functions frequently used in biostatistics [67]. Specifically, one-way analysis of variance and multiple comparisons using Bonferroni correction were performed for the comparison of data across the four groups. For the survival ratio, the data were statistically analyzed using the Kaplan–Meier method with the log-rank test to identify significant differences between groups. *p*-values of <0.05 were considered to be statistically significant.

## Figures and Tables

**Figure 1 ijms-26-01191-f001:**
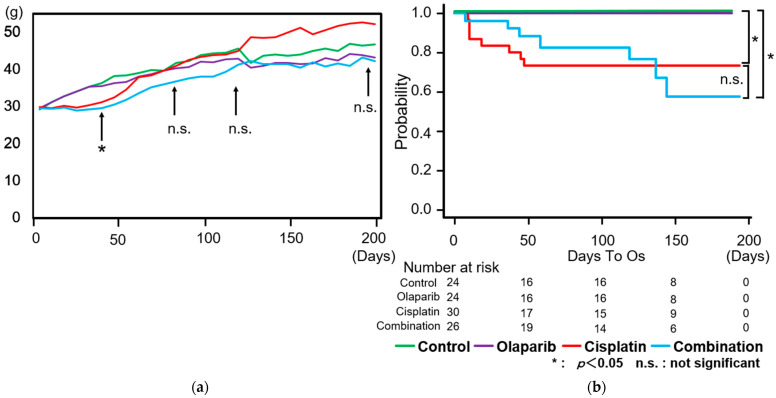
Changes in body weight and survival ratio during the experimental period. (**a**) Body weight changes: body weight increased over time across all groups. One month after drug administration, the cisplatin and combination groups exhibited a significantly lower rate of weight gain compared to another two groups. However, notable recovery in body weight was observed in the cisplatin group, reaching levels of non-significance compared to the control group. (**b**) Survival ratios: survival rates declined significantly in the cisplatin group immediately after drug administration. The combination group showed a gradual decline followed by a notable reduction in survival at 120 days after drug administration. However, this decrease did not reach statistical significance compared to the cisplatin group. By the end of the monitoring period, both the cisplatin and combination groups demonstrated significantly reduced survival rates compared to the other two groups (*p* < 0.05). Data are presented as mean ± SEM. Significance thresholds: *p* < 0.05; n.s., not significant.

**Figure 2 ijms-26-01191-f002:**
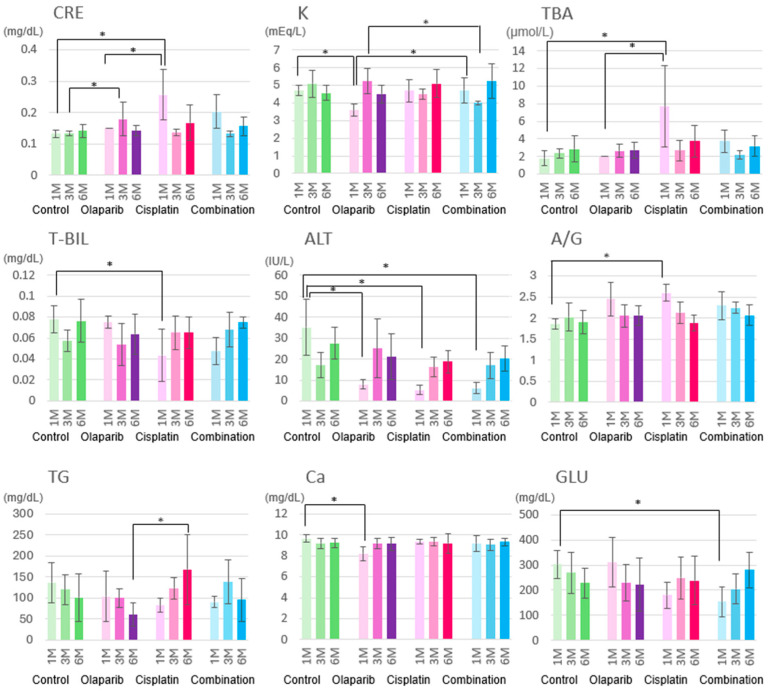
The results of biochemical analysis. At one month after the completion of drug administration, creatinine (CRE) and total bile acid (TBA) levels were significantly increased in the cisplatin group compared to the control and Olaparib groups. Potassium (K) levels showed significant differences between the control and Olaparib group, as well as between the Olaparib and combination groups. Total bilirubin (T-Bil) levels were significantly lower in the cisplatin group compared to control group. Alanine aminotransferase (ALT) levels were significantly reduced in the other three groups compared to the control group. A/G ratio in the cisplatin group was significantly elevated compared to the control group. Triglycerides (TG) were significantly elevated in the cisplatin group six months after the completion of treatment compared to the Olaparib group. Calcium (Ca) levels differed significantly between control and Olaparib groups at one month after completion of drug administration. Glucose levels were significantly lower in the combination group compared to the control group. Statistical significance is indicated as follows: *: *p* < 0.05.

**Figure 3 ijms-26-01191-f003:**
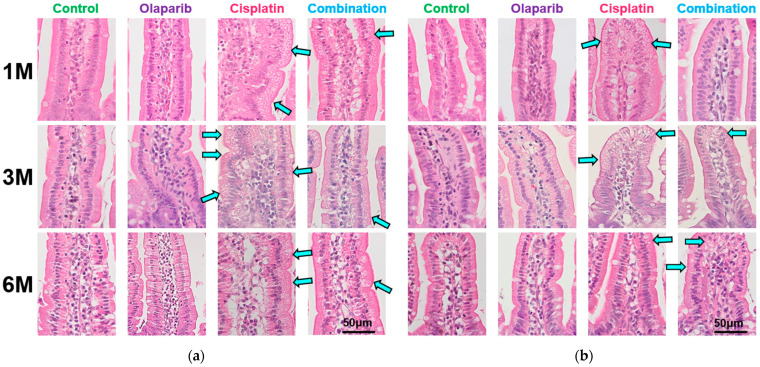
Histopathological assessment of vacuolar degeneration in intestinal villi under higher magnification. (**a**) Pronounced vacuolar degeneration was observed beneath the epithelial surface of jejunal villi in the cisplatin group, with severity followed by the combination group. Arrows highlight areas of severe vacuolar degeneration beneath the epithelial surface. No apparent vacuolar changes were observed in the control or Olaparib groups. (**b**) Similar vacuolar degeneration was observed in ileal villi of the cisplatin and combination groups. Arrows indicate regions of pronounced vacuolar degeneration in the epithelium. The changes rebounded at 6 months in the combination group. Scale bar: 50 μm.

**Figure 4 ijms-26-01191-f004:**
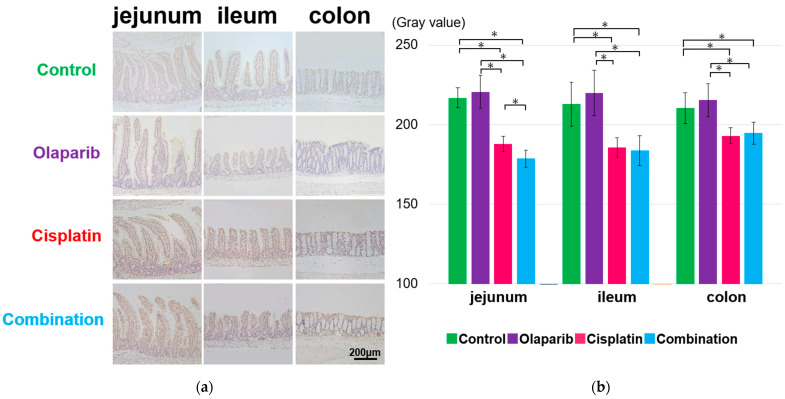
Immunohistochemical analysis of PARP-1 expression at 1 month after drug administration. (**a**) Representative IHC images from each group showing PARP-1 expression in the jejunum, ileum, and colon. Strong PARP-1 positivity was observed not only in the surface mucosa, but also within cellular structures of the villi in the cisplatin and combination groups. Scale bar: 200 μm. (**b**) Semi-quantitative analysis of grayscale values, performed using ImageJ, revealed significantly higher gray values in the control and Olaparib groups, indicating reduced PARP-1 expression. Data are presented as mean ± SEM. *: *p* < 0.05.

**Figure 5 ijms-26-01191-f005:**
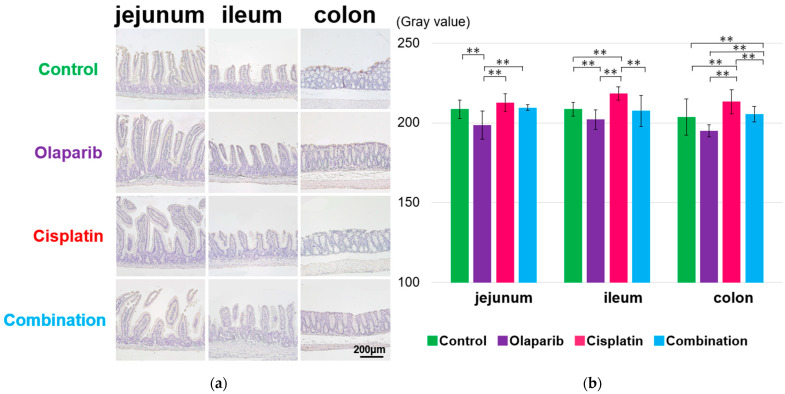
Immunohistochemical analysis of EGF expression in mice after completion of drug administration. (**a**) Representative IHC images from each group showing EGF expression on the epithelial surface of the control and Olaparib groups at 1 month after administration. Scale bar: 200 μm. (**b**) Semi-quantitative grayscale analysis revealed significantly higher gray values in the Cisplatin group at 1 month, indicating reduced EGF expression. Data are presented as mean ± SEM. ** *p* < 0.01.

**Figure 6 ijms-26-01191-f006:**
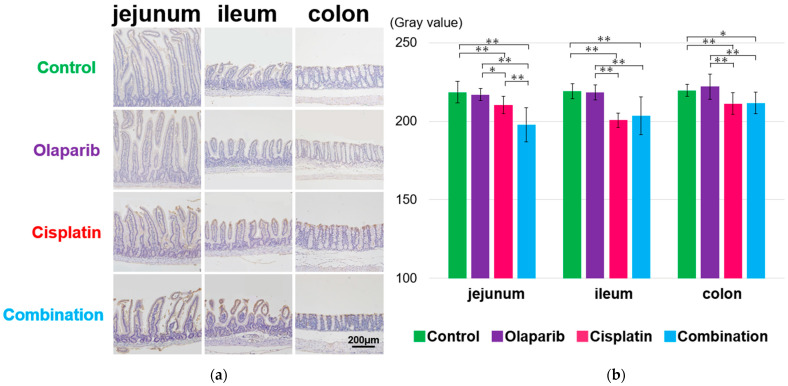
Immunohistochemical analysis of E-cadherin expression in mice after completion of drug administration. (**a**) Representative IHC images from each group showing significantly higher positivity in cisplatin and combination groups. Scale bar: 200 μm. (**b**) Semi-quantitative grayscale analysis revealed significantly lower gray values in the cisplatin and combination group, indicating increased E-cadherin expression. Data are presented as mean ± SEM. * *p* < 0.05, ** *p* < 0.01.

**Figure 7 ijms-26-01191-f007:**
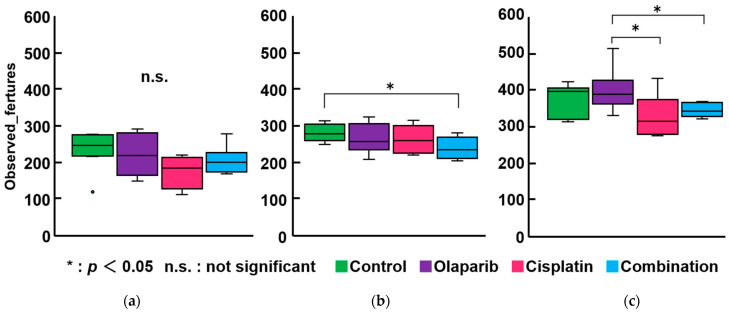
Within-subject α-diversity. (**a**) At 1 month after drug administration, no significant differences were observed among the four groups. (**b**) At 3 months, significant differences in α-diversity were detected between the control and combination groups. (**c**) At 6 months, the Olaparib group exhibited significantly higher α-diversity compared to the cisplatin and combination groups. *: *p* < 0.05; n.s.: not significant.

**Figure 8 ijms-26-01191-f008:**
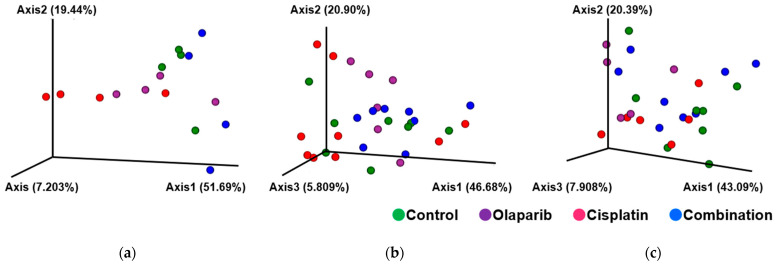
Between-subject β-diversity. (**a**) At one month after drug administration, the cisplatin group exhibited significantly altered β-diversity compared to the control and Olaparib groups. (**b**) At 3 months, the combination group demonstrated significant differences compared to each of the other three groups. (**c**) At 6 months, the cisplatin group showed significantly altered β-diversity compared to Olaparib group.

**Figure 9 ijms-26-01191-f009:**
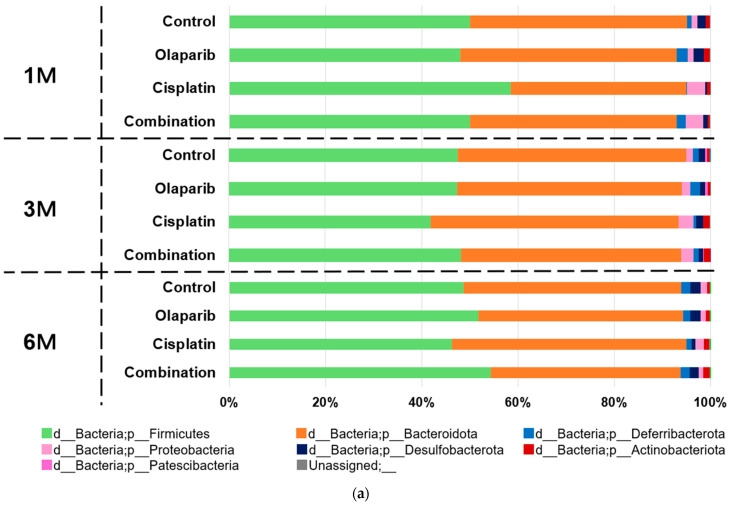
Microbiome composition across the four groups at 1, 3, and 6 months after drug administration. (**a**) Microbiome composition at the phylum level: cisplatin group exhibited significantly higher abundance of Firmicutes and reduced abundance of *Bacteroidata* at 1 month after drug administration. Furthermore, relative increase in *Proteobacteria* was observed in both cisplatin and combination groups at 1 and 3 months after drug administration. (**b**) Microbiome composition at the genus level: *Lachnospiraceae_NK4A136_group* showed significantly higher relative abundance in control and Olaparib groups compared to cisplatin and combination groups at 1 month after drug administration (pink area; *p* < 0.05). Consequently, the combination group exhibited a similar ratio with Olaparib group at 6 months after drug administration. In contrast, *Clostridium_sensu_stricto_1* demonstrated significantly higher abundance in cisplatin group, followed by combination group, compared to the control and Olaparib groups at 1 month (yellow area; *p* < 0.01). Notably, these genus-level differences were no longer significant at 3 and 6 months after drug administration. d: Domain, p: Phylum, c: Class, o; Order, f: Family, g: Genus.

**Figure 10 ijms-26-01191-f010:**
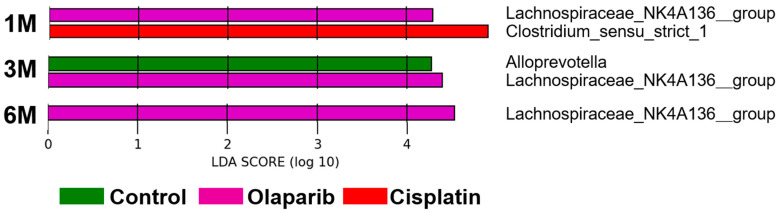
Different abundances microbiota across the four groups identified using LEfSe. Panel represent results at 1, 3, and 6 months after drug administration. Log-transformed LDA scores are plotted on the *x*-axis. LDA scores > 4 were considered statistically significant. Bar length indicates the relative influence of each microbiota.

**Figure 11 ijms-26-01191-f011:**
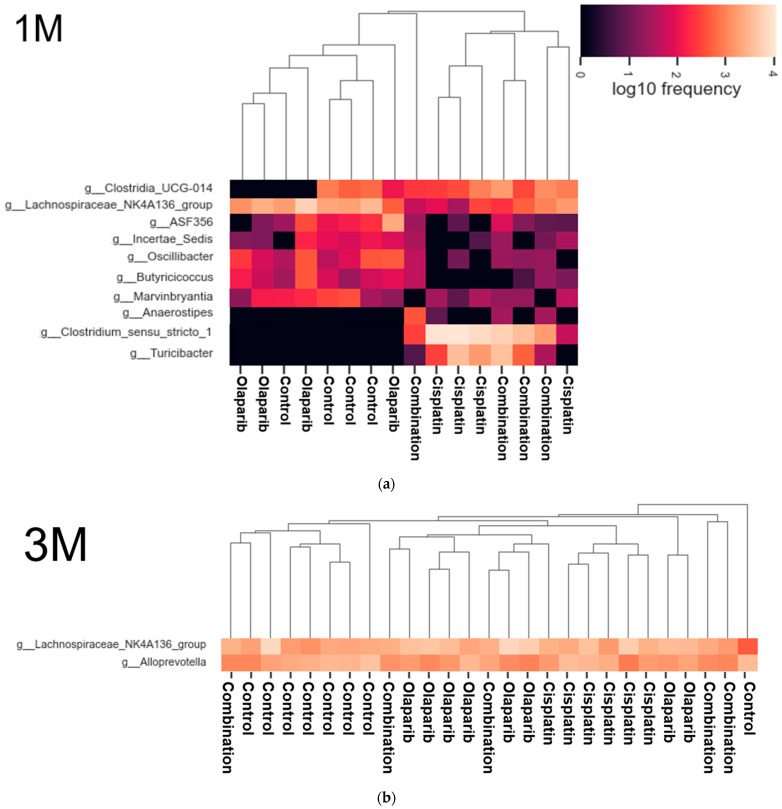
Heat map analysis of changes in intestinal microbiota at the phylum level across the four groups. (**a**) Microbiomes demonstrated significance in abundance 1 month after drug administration was listed. (**b**) Three months after drug administration, no microbiomes showed significant abundance, although g_*Alloprevotella* and g_*Lachnospiraceae_NK4A136_group* showed a high LDA score exceeding 4. (**c**) g_*Turicibacter* and g_*Lachnospiraceae_NK4A136_group* showed significant difference in abundance 6 months after drug administration. A range of colors, from black to beige indicates the relative microbiome value (0–4.0). Microbiomes exhibiting significantly altered relative abundances were identified and extracted from the raw data presented in Appendix A.

**Figure 12 ijms-26-01191-f012:**
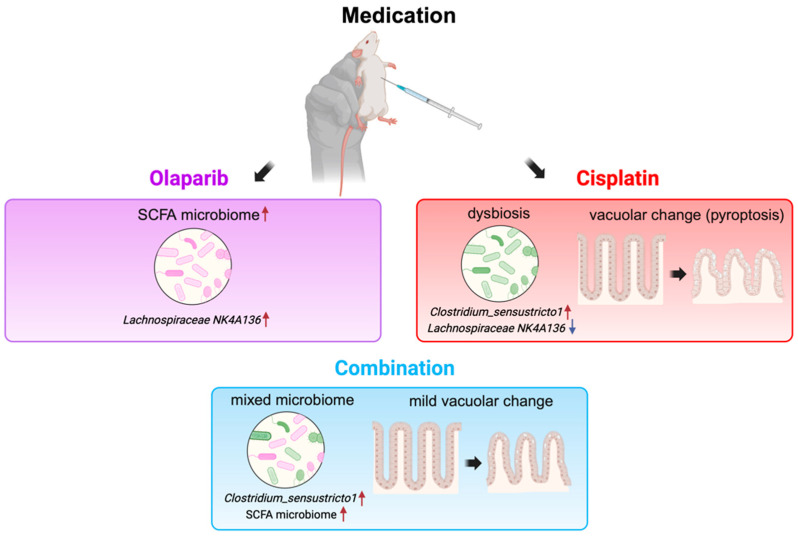
Olaparib exhibited protective effects on intestinal integrity and promoted the abundance of SCFA-producing microbiota, including *NK4A136 group*. In contrast, cisplatin administration resulted in dysbiosis and significant vacuolar changes, indicative of severe cytotoxicity. The combination administration yielded complex and entangled outcomes, potentially attributable to pharmacodynamic interactions involving β-catenin or p53 signaling pathways.

**Table 1 ijms-26-01191-t001:** The number of mice which are analyzed in this study.

	1 M	3 M	6 M
Control Group	8	8	8
Olaparib Group	8	8	8
Cisplatin Group	7	6	9
Combination Group (Both Cisplatin and Olaparib)	8	5	6

## Data Availability

16S rRNA gene amplicon sequence data were registered to Sequence Read Arcive (SRA) on NCBI website (PRJNA1193847).

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
