# Peer review of "Combinatorial Effects of Cisplatin and PARP Inhibitor Olaparib on Survival, Intestinal Integrity, and Microbiome Modulation in Murine Model"

_ijms, 2025, doi:10.3390/ijms26031191_

Round 1
Reviewer 1 Report
Comments and Suggestions for Authors
1. There are too many figures in the main text. It is recommended to streamline them and include unimportant figures in the supplementary materials.
2. There are many short paragraphs in the draft, it is recommended to organize and merge them.
3. The Figure should not present in Discussion Section, such as Line 295 Figure 1b, Line 396 Figure 2, and Line 398 Figure 5.
4. The differences in mortality at different times should be correlated with intestinal differences for analysis. This association analysis can indicate whether the intestine is associated with differences in mortality.
5. The writing logic of this article is somewhat confusing. It is recommended to reorganize the results and discussions to make the article more organized.
Author Response
Thank you very much for your valuable and insightful comments, which have greatly contributed to enhancing the quality of our manuscript. We have thoroughly and carefully revised the manuscript in accordance with your recommendations, with all changes highlighted in yellow. I hope these modifications meet the standards you have requested.
- There are too many figures in the main text. It is recommended to streamline them and include unimportant figures in the supplementary materials.
Reply: We value your insight regarding the presentation of our figures. Following your recommendation, we have made the following adjustments to improve clarity and focus within the main text: former Figures 3, 4, 6(b)(c), and 12 have been relocated to the supplementary materials. Additionally, we re-evaluated the former Figure 13 and 14 (they turned to be Figure 10 and 11 at the revised version), which presents LDA score and the heat-map analysis, respectively, and chose taxa to include figures based on the priority of our manuscript.
- There are many short paragraphs in the draft, it is recommended to organize and merge them.
Reply: Thank you for your insightful comment regarding the organization of our manuscript. In response to your suggestion, we have undertaken a comprehensive revision of the manuscript's structure. We have merged short paragraphs across all sections (Introduction, Results, Discussion, and Material/Methods) to create a more cohesive and engaging narrative. We hope this restructuring has improved the flow of information and enhanced the overall readability of the manuscript.
- The Figure should not present in Discussion Section, such as Line 295 Figure 1b, Line 396 Figure 2, and Line 398 Figure 5.
Reply: Thank you for your insightful comment regarding the placement of figures within the Discussion section. We agree that figures are generally better suited for the Results section where they directly support the findings presented. We have removed the references to all Figures from the Discussion section and ensured that all figures are now appropriately presented only within the Results section. However, we have retained Figure 12 in the Discussion section as it serves as an explanatory figure that aids in the interpretation and synthesis of our findings, which aligns with the journal's encouragement of discussion figures.
- The differences in mortality at different times should be correlated with intestinal differences for analysis. This association analysis can indicate whether the intestine is associated with differences in mortality.
Reply:
Thank you for your insightful comment regarding the factors associated with mortality differences between treatment groups. We appreciate your attention to this important aspect of our study.
We acknowledge that a comprehensive analysis of the relationship between mortality and changes in the intestinal environment would be ideal. However, due to limitations inherent in our study design, we were unable to directly analyze all potential factors associated with mortality.
As you noted, analyzing the intestinal microbiota, blood, and organs immediately after death would provide the most accurate data. However, this presented logistical challenges that precluded such immediate analyses for the mice that died midway through the experiment. Postmortem changes in the intestinal microbiota and blood composition occur rapidly, potentially confounding the results if analyses are not performed immediately. This limitation also restricted us to employ certain statistical methods. Specifically, the lack of data for mice that died during the study prevented us from using the Cox proportional hazards model, which requires data of each factor from each individual with accurate survival time both dropped-out mice and scheduled-analyzed mice. Additionally, our relatively small sample size (6-10 mice per group) made multivariate analysis unsuitable, as it could lead to overfitting and unreliable results. Regarding the data collected at 1, 3, and 6 months after drug administration, we could not apply repeated measures ANOVA because these data points represent different individuals at each time point, violating the assumption of repeated measurements within the same subjects. Furthermore, the unequal sample sizes between groups at each time point pose additional analytical challenges.
Despite these limitations, our analyses at the scheduled time points revealed some potentially relevant findings. In the cisplatin group at 1 month after medication, we observed significant decreases in liver and kidney function, a significant increase in Clostridium_sensu_stricto_1 abundance followed by combination group, and significantly increased expression of PARP in the jejunum. At 3 months, increased PARP expression was also observed in the ileum and colon of the cisplatin group. These changes might contribute to the increased mortality observed in the cisplatin and combination groups, warranting further investigation.
We acknowledge the limitations of our current study and have clearly outlined them in the Discussion section on lines 474-485. We recognize the need for future research with more frequent monitoring and immediate analysis upon death to enable a more comprehensive understanding of the relationship between intestinal microbiome changes, host factors and mortality.
- The writing logic of this article is somewhat confusing. It is recommended to reorganize the results and discussions to make the article more organized.
Reply: Thank you for your valuable feedback regarding the structure and clarity of the Results and Discussion sections. We appreciate your insight and agree that the previous version of the Discussion section could benefit from improved organization and focus.
In response to your comment, we have carefully reviewed and revised the Result and Discussion section.
We have:
- Reorganized the result section and the order for statement were changed
- Removed any sentences that inappropriately reported results rather than interpreting and discussing them and reorganized the remaining sentences to ensure clear and logical flow of ideas.
- Added discussion addressing factors that directly influence mortality, as you suggested in your 4th comment. This includes an acknowledgment of the limitations of our study, specifically our inability to evaluate the factors directly altering mortality.
- As we previously replied, some figures were carefully revised and reorganized based on their priority in our manuscript.
We believe these revisions have significantly improved the clarity and focus of the Discussion section, providing a more insightful interpretation of our findings.
Reviewer 2 Report
Comments and Suggestions for Authors
The authors try to indicate how PARP inhibitor therapy affects the composition of the gut microbiome. Below are some general comments that, in my opinion, must be taken into account before further processing and deeper analysis of this manuscript.
- In the keywords, using Clostridium_Sensu_stricto_1; Lachnospiraceae NK4A136 is too far-reaching a specification, especially since these are detailed names from the database.
- in figures 3 to 8, I suggest starting the Y scale from, for example, 100 to increase the readability of the graphs.
- Figure 9a. What does d_Bacteria mean and how is it possible that in bacteria there is a phylum Archaea and Eukaryota? From what I know, these are completely different kingdoms.
- Figure 9b. The illegible figure suggests reducing the number of types of bacteria presented. In addition, some bacteria are unidentified at the genus level because there are no names, only an underscore. Looking at the diversity, you can see that in two groups there are surprisingly many of the Clostridium_sensu_stricto_1 type, which is probably the result of incorrect sequence assignment. If something like this occurs, I advise the authors to go down to the family level with this type of analysis to avoid it. In this form, the analysis of these data can be very misleading.
- Figure 12 seems to be not very informative and could be included in the supplement.
- Figure 13. The graphs show differences at several taxonomic levels, so in fact they duplicate each other. Let the authors choose a level, for example a family, and re-analyze.
Figure 14. Illegible figures, the authors must limit the number of bacteria included to enable data interpretation.
4.8.3 Bioinformatic analysis
How were the raw data filtered before processing in Qimme? Please provide the average number of reads per sample and the maximum and minimum that were considered.
The authors only performed the analysis of differences via LEfSe. It may also be worth performing univariate analyses between groups and looking for differences between only two groups.
Author Response
The authors try to indicate how PARP inhibitor therapy affects the composition of the gut microbiome. Below are some general comments that, in my opinion, must be taken into account before further processing and deeper analysis of this manuscript.
Thank you very much for your valuable and insightful comments, which have greatly contributed to enhancing the quality of our manuscript. We have thoroughly and carefully revised the manuscript in accordance with your recommendations, with all changes highlighted in blue. I hope these modifications meet the standards you have requested.
- In the keywords, using Clostridium_Sensu_stricto_1; Lachnospiraceae NK4A136 is too far-reaching a specification, especially since these are detailed names from the database.
Reply: Thank you for your insightful comment regarding the keywords. We agree that "Clostridium_Sensu_stricto_1" and "Lachnospiraceae NK4A136" are overly specific and might not be readily recognizable to a broader audience within the field. As these terms reflect detailed classifications from the database, we have removed them from the keywords to ensure clarity and improve accessibility for readers. We have replaced them with "dysbiosis" and "microbial metabolism" to better capture the essence of the research.
- in figures 3 to 8, I suggest starting the Y scale from, for example, 100 to increase the readability of the graphs.
Reply: Thank you for this helpful suggestion. We agree that adjusting the Y-axis scale will significantly improve the clarity and readability of previous Figures 3 to 8, which turned to be Figure 4-6, S1 and S2 of revised version. As you recommended, we have modified the graphs to start the Y-axis at 100. This change allows for better visualization of the data trends and subtle differences between the groups, enhancing the overall impact and interpretability of the figures.
- Figure 9a. What does d_Bacteria mean and how is it possible that in bacteria there is a phylum Archaea and Eukaryota? From what I know, these are completely different kingdoms.
Reply: Thank you for your comment regarding former Figure 9a. We understand your concern about the presence of 'Archaea' and 'Eukaryota' within the 'Bacteria' category. To clarify, 'd_Bacteria' in our figures refers to the domain Bacteria, the highest taxonomic rank, encompassing all bacteria. Our analysis with Qiime 2 initially classifies sequences into the three domains: Bacteria, Archaea, and Eukaryota. While a small proportion of archaeal and eukaryotic sequences were detected, these were not statistically significant and did not meaningfully contribute to the overall microbial composition. Therefore, for clarity and conciseness in Figure 9a, we focused on the bacterial phyla within the domain Bacteria. The archaeal and eukaryotic sequences are included in the "Unassigned" area, represented by the gray color in the figure.
We apologize if the figure labeling caused any confusion. We will revise the figure caption of Figure 9 to explicitly state that 'd_Bacteria' represents the domain Bacteria to avoid any further misunderstanding.
- Figure 9b. The illegible figure suggests reducing the number of types of bacteria presented. In addition, some bacteria are unidentified at the genus level because there are no names, only an underscore. Looking at the diversity, you can see that in two groups there are surprisingly many of the Clostridium_sensu_stricto_1 type, which is probably the result of incorrect sequence assignment. If something like this occurs, I advise the authors to go down to the family level with this type of analysis to avoid it. In this form, the analysis of these data can be very misleading.
Reply: Thank you for your insightful comments on Figure 9b. We appreciate your attention to detail and understand your concerns about the figure's readability and the potential for misinterpretation of the data.
- Improved figure design:
- We have added dotted lines to delineate the different time points, making it easier to distinguish between groups and track changes over time.
- The number of taxa displayed has been reduced, focusing on those most relevant to the study's objectives, to improve readability and highlight key findings.
- Handling of unidentified taxa:
Taxa that were not identified at the genus level have been removed from the figure. This adjustment was necessary due to limitations in data processing using Qiime 2 software. Although the taxa were classified to level 6, some were not resolved to the genus level, and manual intervention was required to exclude these unidentified taxa. We have ensured that the remaining taxa presented are reliable and relevant for interpretation.
- Re-evaluation of taxonomic assignments:
Regarding the abundance of Clostridium_sensu_stricto_1, we recognize the importance of this observation and its potential implications for our study. We have carefully re-examined the taxonomic classification of these sequences and confirmed their assignment to Clostridium_sensu_stricto_1 with high confidence. Although we acknowledge the inherent limitations of 16S rRNA gene sequencing, particularly at the species level, we are confident that the observed abundance changes accurately reflect the effects of cisplatin and combination therapy on the intestinal microbiome.
We also appreciate your suggestion to analyze the data at the family level rather than the genus level. While we understand the potential advantages of this approach, the primary objective of our study was to identify specific bacteria significantly affected by PARP inhibitor monotherapy. For instance, we identified the bacterium NK4A136 at the genus level as significantly increased. Furthermore, the LDA score for Clostridium_sensu_stricto_1 was greater than 4 at both the order and family levels, as shown in the former Figure 13 (Figure 10 at the revised version), supporting the increase of Clostridium_sensu_stricto_1 described in Figure 9b. We also modified Figure 13 (Figure 10 at the revised version) in accordance with your 6th comments, and we will state it at the reply to the 6th comments.
- Figure 12 seems to be not very informative and could be included in the supplement.
Reply: Thank you for your comment regarding Figure 12. As both reviewers suggested moving this figure to the supplement, we have transferred Figure 12 (LEfSe Analysis) to the Supplementary Materials.
- Figure 13. The graphs show differences at several taxonomic levels, so in fact they duplicate each other. Let the authors choose a level, for example a family, and re-analyze.
Reply: Thank you for your insightful comment regarding Figure 13. We agree that presenting the same taxa at multiple taxonomic levels in the original figure was redundant and could lead to confusion. To address this, we have revised the figure to focus on the genus level, which we believe provides the most relevant and specific information for this analysis. While the initial presentation was based on the understanding that displaying multiple levels could illustrate the robustness of the LDA score across those levels, we recognize that this approach compromised clarity. The revised figure (now Figure 10) presents the LDA scores at the genus level, highlighting the key findings without unnecessary repetition. This change enhances the figure's clarity and ensures a more concise presentation of the results."
- Figure 14. Illegible figures, the authors must limit the number of bacteria included to enable data interpretation.
Thank you for your comment regarding the clarity of Figure 14 (which turned to be Figure 11 of revised version). We agree that the original figure contained an excessive number of taxa, which hindered data interpretation. To improve readability and focus on the most relevant findings, we have revised Figure 14 by removing taxa with less priority.
- 8.3 Bioinformatic analysis
How were the raw data filtered before processing in Qimme? Please provide the average number of reads per sample and the maximum and minimum that were considered.
The authors only performed the analysis of differences via LEfSe. It may also be worth performing univariate analyses between groups and looking for differences between only two groups.
Reply: Thank you for your valuable feedback regarding the bioinformatic analysis section. We apologize if the previous presentation was unclear, and we appreciate your attention to this detail.
To clarify the data processing steps, we have provided additional information regarding the raw sequence data on line 237-242 (result section) and 673-678 (material and method section).
- Sequencing Depth: The raw data received from Genome-Lead Co. Ltd. consisted of both forward and reverse reads. The minimum number of reads per sample was 57,205 bp, the mean number of reads was 95,680.89 bp, and the maximum number of reads was 137,920 bp.
- Cluster Density and Phix Control Reads: The resulting data showed a cluster density of 775K/mm² with 16.34% Phix control reads.
- Quality Control: A high percentage of reads passed quality filtering (94.82%), indicating good sequencing quality. The Q30 score was 82.87%, demonstrating high accuracy in base calling.
These metrics confirm that the raw data quality is suitable for further microbial analysis.
Regarding the analysis methods, we used QIIME2 v2021.4 to conduct several analyses beyond LEfSe. Here's a summary:
- Alpha Diversity: We assessed within-subject alpha diversity using the Shannon index and the observed number of operational taxonomic units (OTUs). These indices were compared among the four groups using Kruskal–Wallis test.
- Beta Diversity: We evaluated between-subjects beta diversity using Bray–Curtis dissimilarity and unweighted and weighted UniFrac distance metrics. Principal coordinate analysis (PCoA) was performed to visualize global differences in microbiome structure. Statistical Analysis was performed by PERMANOVA. Alpha and Beta diversity were performed using Qiime 2.
- Microbiome composition analysis at level 2 and 6
- Analysis of Composition of Microbiomes (ANCOM)
To further explore differences in microbiome composition at level 2 and 6, we conducted the one-way analysis of variance with bonferroni correction for each taxa at level 2 and 6 using EZR software. While the software automatically performs univariate analysis, we acknowledge the limitations of these comparisons without appropriate adjustments for multiple comparisons. Therefore, we primarily rely on the one-way analysis of variance with bonferroni correction to assess overall differences in taxa abundance among the four groups. We believe this approach provides a more robust assessment of the overall patterns in microbiome composition. Moreover, we additionally performed ANCOM analysis, which support the result of g_Lachnospiraceae_NK4A136_group to be specific and significantly different abundance (Lines 360-364, 700-707).
Round 2
Reviewer 2 Report
Comments and Suggestions for Authors
Figure 9a still shows Archaea and Eukaryota.
I understand that the presence and change in the proportions of the Clostridium_sensu_stricto species is a key aspect in this paper. A large part of the conclusions in the paper are based on this finding and I know from experience that the analysis of 16s and the assignment of species may be incorrect, so if the authors want to rely on this finding here and go down to the species level, in my opinion they need to confirm this finding by another method, for example the Real-time technique. Relying only on 16s is too unreliable in my opinion.
Author Response
Thank you very much for your valuable and insightful comments, which have further contributed to enhancing the quality of our manuscript. We have thoroughly and carefully revised the manuscript in accordance with your comments. I hope these modifications meet the standards you have requested.
1 Figure 9a still shows Archaea and Eukaryota.
Thank you very much for your comments. We have carefully re-checked the data and updated it.
2 I understand that the presence and change in the proportions of the Clostridium_sensu_stricto species is a key aspect in this paper. A large part of the conclusions in the paper are based on this finding and I know from experience that the analysis of 16s and the assignment of species may be incorrect, so if the authors want to rely on this finding here and go down to the species level, in my opinion they need to confirm this finding by another method, for example the Real-time technique. Relying only on 16s is too unreliable in my opinion.
Reply:
Thank you very much for your thoughtful comments regarding the 16S rRNA microbiome analysis in our study. We appreciate your comments of the reliability of the 16S rRNA microbiome analysis, offer of species-level assignments and the potential value of incorporating additional methods, such as real-time PCR, to corroborate our findings.
As you have noted, real-time PCR offers superior accuracy for targeted quantification of specific microbial taxa, whereas 16S rRNA sequencing provides a comprehensive overview of microbial community composition. We fully agree that real-time PCR would strengthen the evidence for the presence of 16S rRNA-detected taxa such as Clostridium sensu stricto-1 and could further substantiate our conclusions.
However, we would like to highlight the context of our current study. While this study aimed to provide a broad overview of the microbial community, we acknowledge its limitations, including the lack of real-time PCR validation. To address this, follow-up investigations are already underway from last September as a part of the thesis project of one of our co-authors, Mr. Sakai, who is also a graduate student at Tsurumi University. His ongoing work focuses on the detailed analysis of ”Clostridium sensu stricto-1“ and ”NK4A136”, both of them are identified and featured in Ms. Matsumura’s current findings, exploring their associations with short-chain fatty acids (SCFA), inflammation, condition of another organs (liver, kidney, and spleen) and hematological assessments using the same murine model. Considering these circumstances, we’d like to conclude the microbiome analysis for this manuscript at the broader community level, and conduct subsequent studies aimed at addressing the finer details, including real-time PCR validation as subsequent research. We believe this approach provides a new starting point for more focused investigations in the future while acknowledging the study’s limitations. We deeply appreciate your insight and hope this explanation clarifies the scope and direction of our work.
Furthermore, we apologize if our description appeared to emphasize the microbiome analysis more than intended. To clarify, while our study does include microbiome analysis notably (5 out of the 12 main figures, 2 supplemental tables, and 6 supplemental figures), we have also devoted significant focus to histopathological analysis. Specifically, we utilized high magnification with a 60x objective lens for detailed examination (Figure 3) and performed immunohistopathological analysis using three antibodies, which are represented in 4 out of the 12 main figures, Tables S1-3, and 7 supplemental figures.
As previously stated, the findings from the microbiome analysis have inspired subsequent research, which has already secured a new grant from the Ministry of Education and Science of our country and Mr. Sakai has already started his thesis project. These results serve as a foundation for ongoing investigations, but we have taken steps to ensure a balanced presentation of both the histopathological and microbiome aspects of the study. To reflect this, we have revised the title and keywords as follows, discussion (lines 484-488), and conclusion (lines 494-497) to adopt a more neutral tone, ensuring equal emphasis on both aspects of the research. All revised sections are highlighted in with bold letter and underline for your convenience.
Title;
Previous: Combinatorial effects of cisplatin and PARP inhibitor demonstrated decreased side effects on intestinal microbiome for post-administrating evaluation
Changed: Combinatorial effects of cisplatin and PARP inhibitor Olaparib on Survival, Intestinal Integrity, and Microbiome Modulation in a Murine Model
Key words;
Previous: PARP inhibitor; Olaparib; Cisplatin; microbiome; 16S rRNA Sequencing; dysbiosis; microbial metabolism
Changed: PARP inhibitor; Olaparib; Cisplatin; microbiome; pyroptosis; survival; dysbiosis;
We are sincerely grateful for your feedback, which has helped us refine our manuscript to align with the high standards of your journal. We kindly request your reconsideration of these changes.